# Effects of Whey Protein Supplementation on Aortic Stiffness, Cerebral Blood Flow, and Cognitive Function in Community-Dwelling Older Adults: Findings from the ANCHORS A-WHEY Clinical Trial

**DOI:** 10.3390/nu12041054

**Published:** 2020-04-10

**Authors:** Wesley K. Lefferts, Jacqueline A. Augustine, Nicole L. Spartano, William E. Hughes, Matthew C. Babcock, Brigid K. Heenan, Kevin S. Heffernan

**Affiliations:** Human Performance Laboratory, Department of Exercise Science, Syracuse University, Syracuse, NY 13244, USA; wleffert@uic.edu (W.K.L.); jacqueline.augustin@cortland.edu (J.A.A.); spartano@bu.edu (N.L.S.); whughes@mcw.edu (W.E.H.); Matthew.Babcock@CUAnschutz.EDU (M.C.B.); brigid.heenan@gmail.com (B.K.H.)

**Keywords:** vascular stiffness, blood pressure, whey protein isolate, older adults

## Abstract

ANCHORS A-WHEY was a 12-week randomized controlled trial (RCT) designed to examine the effect of whey protein on large artery stiffness, cerebrovascular responses to cognitive activity and cognitive function in older adults. Methods: 99 older adults (mean ± SD; age 67 ± 6 years, BMI 27.2 ± 4.7kg/m^2^, 45% female) were randomly assigned to 50g/daily of whey protein isolate (WPI) or an iso-caloric carbohydrate (CHO) control for 12 weeks (NCT01956994). Aortic stiffness was determined as carotid-femoral pulse wave velocity (cfPWV). Aortic hemodynamic load was assessed as the product of aortic systolic blood pressure and heart rate (Ao SBP × HR). Cerebrovascular response to cognitive activity was assessed as change in middle-cerebral artery (MCA) blood velocity pulsatility index (PI) during a cognitive perturbation (Stroop task). Cognitive function was assessed using a computerized neurocognitive battery. Results: cfPWV increased slightly in CHO and significantly decreased in WPI (*p* < 0.05). Ao SBP × HR was unaltered in CHO but decreased significantly in WPI (*p* < 0.05). Although emotion recognition selectively improved with WPI (*p* < 0.05), WPI had no effect on other domains of cognitive function or MCA PI response to cognitive activity (*p* > 0.05 for all). Conclusions: Compared to CHO, WPI supplementation results in favorable reductions in aortic stiffness and aortic hemodynamic load with limited effects on cognitive function and cerebrovascular function in community-dwelling older adults.

## 1. Introduction

Vascular dysfunction is a phenotypic expression of human aging that contributes to increases in cardiovascular and cerebrovascular disease prevalence in older adults [1]. With advancing age, central elastic arteries become stiffer, owing to numerous structural and functional aberrations [2]. This stiffening impairs the inherent buffering capacity of the large central arteries and increases blood pressure (BP) and blood flow pulsatility. Arterial stiffness and subsequent increases in central hemodynamic pulsatility are associated with several pathologies of aging including hypertension, left ventricular hypertrophy and heart failure, renal dysfunction, and retinal damage [2]. Moreover, increased arterial stiffness and central hemodynamic pulsatility are independent predictors of cardiovascular and cerebrovascular events and mortality [3,4]. 

While prolonging life is an important public health goal, preserving the capacity to live and function independently is equally significant [5]. Identifying proven interventions that can prevent disability is a major public health challenge. Cognition is an important contributing factor to overall functional ability and quality of life with advancing age [6,7]. The brain is a high flow target organ that is particularly sensitive to excessive hemodynamic pulsatility, with central hemodynamic pulsatility potentially infiltrating and damaging the delicate cerebral microvasculature. Numerous studies note relationships between central artery stiffness, pulsatile hemodynamics, cerebrovascular pulsatility, and cognitive function [8,9,10]. Arterial stiffness and cerebral pulsatility also predict cognitive decline with advancing age and incident dementia [11,12]. As such, the American Heart Association and the American Stroke Association acknowledge the importance of arterial stiffness as a significant factor governing cognitive impairment with aging and disease, advocating early intervention to postpone or prevent onset of vascular cognitive impairment [13]. 

Interventions that improve vascular function may in turn have favorable effects on cognitive function. Combining nutrition with pharmacology has given rise to nutraceuticals: foods and/or dietary supplements with bioactive properties leading to possible physiological and health benefits. Whey protein is one such nutraceutical with the potential to improve both cardiovascular and cognitive health [14,15]. Whey protein comprises approximately 20% of the protein in milk. Milk proteins like whey may be one of the mechanisms partially responsible for associations between higher dairy consumption and reduced risk for incident hypertension [16], reduced arterial stiffness [17] and improved cognitive function [18]. Indeed, whey protein is encrypted with angiotensin converting enzyme (ACE) inhibitory peptides (i.e., lactokinins) [19] and is a notable source of l-arginine, the precursor for nitric oxide. Acute whey protein intake is associated with improved vascular endothelial function [20] and cognitive function [21] while longer term (12 weeks) consumption has been shown to improve endothelial function, lower brachial blood pressure and lower central hemodynamic load (assessed as augmentation index or blood pressure attributable to global wave reflections) [22,23]. Considering arterial stiffness may represent a novel modifiable target for cognitive impairment in older adults, whey protein supplementation could serve as a beneficial nutraceutical strategy to stave both cardiovascular and cognitive decline in older adults. 

The Aging, Neurocognitive, and Cardiovascular Health Outcomes Research Study: Add Whey (ANCHORS A-WHEY) was a double-blind, placebo controlled, randomized controlled trial designed to compare the effects of whey protein isolate (WPI) supplementation to a carbohydrate (CHO) control on large artery stiffness (aortic and carotid), central blood pressure pulsatility, cerebrovascular response to cognitive activity, and cognitive function in community-dwelling older adults. We hypothesized that compared to CHO, WPI would: (1) lower large artery stiffness and central blood pressure pulsatility; (2) improve the cerebrovascular response to cognitive activity; (3) improve cognitive function. 

## 2. Methods

This study was approved by the Institutional Review Board of Syracuse University and all participants were required to provide written informed consent prior to study initiation. One-hundred and twenty-two men and women between 60–85 years of age voluntarily participated in this study. Participants were recruited from the community via local newspaper and radio advertisements. Exclusion criteria included self-reported history of stroke, Alzheimer’s disease, neurological disease of any kind, smoking, head trauma (i.e., concussion/loss of consciousness within the past 6 months), diabetes mellitus, pulmonary disease, severe arrhythmia, peripheral artery disease, renal disease, habitual consumption of whey protein supplements, and laboratory measured severe obesity (body mass index ≥35 kg/m^2^), high depressive symptomology (score >18), cognitive impairment (Montreal Cognitive Assessment score <24), and color blindness. This study was registered at ClinicalTrials.gov (NCT01956994).

Participants were randomized to supplement their regular, daily diet with 50 g of WPI (NOW food brands whey protein isolate) or 50 g of carbohydrate (NOW Foods Brand Carbogain, maltodextrin) as an iso-caloric control condition. The randomization scheme was generated by the study coordinator using an online resource (randomization.com). Participants were instructed to consume two servings of 25 g (50 g/day) for 12 weeks. This dosage and study length were chosen based on a previous study noting changes in blood pressure and vascular function with similar dosages and similar length interventions [23]. Both supplements were approximately 100 kcals per serving and had <0.5 g fat per serving. Supplements were distributed in powder form in pre-measured packets. Both supplements were Vanilla flavored and similar in color and composition to ensure both participants and research study personnel were blinded to condition. Participants were instructed to maintain their habitual physical activity and diet during the intervention trial but refrain from consuming additional protein supplements. Supplements were given in 3-week supply. Participants returned to the Human Performance Lab every 3-weeks to receive a new supply, complete a urine test to assess global kidney function (urinary protein, glucose, ketones, leukocytes, nitrite, pH, specific gravity and P:C ratio) and return empty packets as a qualitative assessment of compliance.

### 2.1. Study Design

At the study onset, participants reported to the Human Performance Laboratory for two separate visits. Each visit occurred first thing in the morning (0600-0900) after an overnight fast. For the consent and initial screening visit, participants completed a health history questionnaire, visual acuity and Ishihara color-blindness tests, basic body anthropometrics (height, weight and waist circumference), and depressive symptomology (Center for Epidemiologic Studies Depression Scale (CES-D)) and global cognitive function assessment (Montreal Cognitive Assessment (MOCA)). Additional assessments included body composition via air displacement plethysmography (BodPod; COSMED, Concord, CA), urinalysis to assess the presence/absence of glucose, protein and ketones, and creatinine levels in the urine (Clinitek Status+ Analyzer, Siemans, IL), and fasting glucose and lipid levels via finger stick (Cholestech LDX). Participants were familiarized with all instrumentation and vascular-hemodynamic measures. Participants also completed a 6m walk test to assess gait speed as a measure of global physical function.

Participants returned to the Human Performance Lab approximately 7 days after the initial screening visit for vascular and hemodynamic data acquisition. Vascular-hemodynamic testing occurred in a quiet, dimly lit, temperature-controlled laboratory during the morning hours following an overnight fast. Participants were instructed to abstain from vigorous exercise and caffeine/alcohol consumption for ≥12 h before testing. Participants did not refrain from taking essential medication. Participants were once again familiarized with all instrumentation and vascular-hemodynamic measures. Following familiarization, participants were instrumented and rested in the supine position for 15 min. Participants remained supine for all hemodynamic measures. Baseline measures were then collected for blood pressure (brachial and aortic), blood flow velocity (carotid and cerebral), and large artery stiffness (carotid and aortic). Select hemodynamic measures were then reassessed (in duplicate) during a 4-min cognitive perturbation protocol to assess cerebrovascular responses to cognitive activity (described below). Finally, participants completed a 30-min computerized cognitive testing battery. All aforementioned measures were again completed following the 12-week intervention in a single visit in the morning (within 48 h of consuming the last dose of WPI/CHO). 

### 2.2. Brachial Blood Pressure

Blood pressures were measured using a validated, automated oscillometric cuff (EW3109, Panasonic Electric Works, Secaucus, NJ, USA). Blood pressure readings were taken in duplicate, with additional readings acquired if values differed by >5 mmHg. Mean and pulse pressure were calculated as 1/3 systolic pressure + 2/3 diastolic pressure, and systolic pressure – diastolic pressure, respectively. 

### 2.3. Aortic Stiffness and Blood Pressure

Pressure waveforms from the right carotid and right femoral arteries were acquired via applanation tonometry. Following palpation of the carotid artery pulse and femoral artery pulse, the distance (in mm) between the carotid artery pulse site and suprasternal notch and femoral artery pulse site and suprasternal notch was obtained with a tape measure. Aortic path length (transit distance) was estimated as suprasternal notch-carotid distance subtracted from the suprasternal notch-femoral distance. Carotid-femoral pulse wave velocity (cfPWV) was calculated as the transit distance/transit time. All measures followed current professional consensus recommendations [2]. 

Applanation tonometry was also used to capture two 10-s epochs of radial artery pressure waveforms (Millar Instruments, Houston, TX, USA). A single composite central aortic pressure waveform was reconstructed from the aforementioned radial artery pressure waveforms using a generalized validated transfer function, (SphygmoCor, AtCor Medical, Sydney, NSW, Australia). The synthesized aortic pressure waveforms were calibrated to brachial mean and diastolic pressure as mean and diastolic pressure are assumed to be somewhat stable throughout the systemic circulation. Augmentation index (AIx) was calculated as the ratio of amplitude of the pressure wave above its systolic shoulder to the total pulse pressure expressed as a percentage ((*P*_2_ − *P*_1_)/PP × 100). AIx was also expressed relative to a standardized heart rate as AIx@75. Aortic rate pressure product was calculated as aortic systolic pressure x heart rate and taken as a measure of central hemodynamic load. Measures were made in duplicate and averaged values used for subsequent analyses. If AIx differed between measures by >5% suggesting a difference in pulse wave contour, a third measure was taken and the average of the 2 closest measures used for analyses. 

### 2.4. Carotid Blood Flow and Stiffness

The left common carotid artery (CCA) was imaged 5–7 cm below the carotid bulb using Doppler ultrasound (ProSound α7, Aloka, Tokyo, Japan) and 7.5–10.0 mHz linear-array probe. The distance between the near wall and far wall lumen-intima interface was continuously traced with eTracking software and used to assess maximum (systolic) and minimum (diastolic) CCA diameters (determined from simultaneous ECG gating from a single lead modified CM5 configuration). Blood velocity waveforms were measured using range gated color Doppler signals averaged along the Doppler beam. Insonation angles were maintained at ≤60° all measures, with sample volume manually adjusted to encompass the entire vessel. CCA pulsatility index (PI) was calculated using semi-automated flow tracing software as (*V*_s_ − *V*_d_)/*V_m_*, where vs. is the peak systolic velocity, *V*_d_ diastolic velocity and *V_m_* the mean velocity.

CCA β-stiffness was determined as ln(P_max_/P_min_)/(D_max_ − D_min_)/D_min_), where P and D correspond to carotid pressure and diameter, respectively, and Max and Min refer to the maximum (systolic) and minimum (diastolic) values during the cardiac cycle. Carotid pressure was simultaneously obtained from the right carotid artery via applanation tonometry from a 10 s epoch (SphygmoCor, AtCor Medical, Sydney, NSW, Australia). Carotid pressure waveforms were calibrated in the same manner as the synthesized aortic pressure waveform, described above.

CCA wave intensity was calculated using time derivatives of blood pressure (P) and velocity (U), where wave intensity = (dP/dt × dU/dt); thus the area under the dP/dt × dU/dt curve represents the energy transfer of the wave. According to WIA, W_1_ characterizes a forward compression wave generated by left ventricular contraction that accelerates flow and increases pressure; the negative area (NA) occurring after W_1_ is a backward travelling compression wave (wave reflection) that decelerates flow but augments pressure. CCA WIA was measured to provide insight into cerebrovascular function as changes in NA in the CCA are thought to be due to wave reflections from cerebral origin [24] and changes in CCA WIA predict cognitive decline in later-life [25]. 

### 2.5. Cerebral Blood Flow Velocity

Left middle cerebral artery (MCA) blood velocity was measured using a 2-mHz transcranial Doppler ultrasound probe (DWL Doppler Box-X, Compumedics, Germany) applied to the temporal window. Mean MCA blood velocity and PI were measured at depths of 45–60 mm, as has been commonly reported for MCA measurements. Mean velocity was calculated from the velocity spectrum envelope using a standard algorithm implemented on the instrument with use of a fast Fourier transform. MCA pulsatility index was calculated with automated flow tracing software using the same equation as defined previously for CCA PI.

### 2.6. Cerebrovascular Response to Cognitive Activity

Participants remained supine while a specialized wall mount suspended a 42-inch flat screen television horizontally over the participant. The television interfaced with a laptop (Dell) and remote response clicker to run a 4-min customized color-word interference Stroop task (E-Prime 2.0, Psychology Software Tools Inc., Sharpsburg, PA, USA). A detailed description of this protocol may be found here [26,27]. This cognitive task has been used previously to assess cardiovascular responses and neural activation to cognitive stimuli during fMRI [28]. Brachial blood pressure, CCA diameter and MCA blood velocity were each measured in duplicate during the Stroop task. We operationally defined cerebrovascular responses to cognitive activity as the change from rest to Stroop for: (1) CCA diameter; (2) MCA mean velocity; (3) and MCA PI. Change in CCA diameter during mental stress has previously been used as a measure of carotid endothelial function [29]. Additionally, change in MCA PI during mental stress has previously been used as a measure of neurovascular coupling and been shown to predict cognitive performance in older adults [30]. 

### 2.7. Computerized Cognitive Function Battery

All participants completed a comprehensive computerized neurocognitive battery that interrogated numerous cognitive domains including executive function, attention, information processing, response speed/sensorimotor function, impulsivity, memory, and emotion recognition (social cognition). For a detailed description of the tasks, please see our previous work [31].

### 2.8. Physical Activity

Physical activity was assessed qualitatively via the short form International Physical Activity Questionnaire (IPAQ), and quantitatively via accelerometry (ActiGraph GT3X+ accelerometer; ActiGraph LLC, Pensacola, FL, USA) in a subset of participants (WPI *n* = 34, CHO *n* = 32). This was done to ensure no seasonal changes in physical activity across the duration of the intervention as a potential confounder of vascular and cognitive function. Accelerometers were worn on the waist (directly below the right mid-axillary line) for 7 consecutive days. Data from the GT3X+ device were downloaded using the low frequency filter from the ActiLife software (version 6.13, ActiGraph LLC, Pensacola, FL, USA). Participants needed to acquire a minimum of 4 days of wear data with at least 10 h of awake wear time per day to be included in data analysis [32]. Raw accelerometer data was converted to counts and summed over a 60 sec epoch for days that accrued at least 10 h of awake wear time. Furthermore, periods of non-wear were defined as consecutive blocks of at least 60 min of 0 activity counts, including up to 2 consecutive minutes of activity counts less than 100, in line with the National Health and Nutrition Examination Survey (NHANES) criteria [32]. A cut point of 2020 activity counts/min was used to determine the amount of time in minutes spent at a physical activity level of moderate-to-vigorous intensity (MVPA) [32].

### 2.9. Statistical Approach

#### 2.9.1. Sample Size Estimation

Sample size estimates were based on the anticipated differences and standard deviation in large artery stiffness between the WPI and CHO group since we hypothesized that all changes in cerebral and cognitive function would stem from changes in large artery stiffness. Samples sizes were estimated by using R.V. Lenth’s Java Applets for Power and Sample Size (retrieved April 9, 2012, from http://www.stat.uiowa.edu/~rlenth/Power). Sample sizes were calculated to give 80% power for an α 0.05 level, two-sided test. Studies note an approximate 20%–30% reduction in large artery stiffness (increase in compliance) following various pharmacological, dietary and other lifestyle interventions of similar length [33,34,35,36,37]. This yields an effect size f as ranging from approximately 0.35–0.49. Thus, for a power of 0.8 with an alpha set at 0.05 for a two tailed test, approximately 30–40 participants per group would be needed to detect a similarly-sized main effect in central artery stiffness. Based on the sample size estimates from our primary outcomes, an estimated drop-out rate of ~10%, and poor transcranial Doppler windows in ~10% of older adults precluding measurement of MCA flow velocity, we enrolled 120 participants with the goal of 40 subjects per group completing this RCT to detect significant changes in desired outcomes.

#### 2.9.2. Statistical Analyses

All data are reported as mean ± standard deviation with statistical significance established a priori as *p* < 0.05. Data normality was assessed quantitatively using the Shapiro-Wilk test, with non-normal data logarithmically transformed to meet normality assumptions. Descriptive characteristics between WPI and CHO groups were compared using independent T-tests for continuous variables and χ^2^ tests for categorical data. 

The effects of 12-week supplementation with WPI compared to CHO on using vascular and secondary outcomes (body weight, lipids, physical activity) were examined using a 2 × 2 (2 group × 2 time) repeated measures ANOVA. Any significant group by time interactions were further explored using Bonferroni corrected post-hoc tests. Cognitive performance outcomes were non-normally distributed and unable to be successfully transformed to meet normality assumptions. Cognitive function and physical activity (IPAQ, MVPA) metrics were thus analyzed via Mann–Whitney U-tests to test the effect of group (WPI vs. CHO), and group by time interaction (change in cognitive function metric post-pre for WPI vs. CHO), with Wilcoxon signed-rank tests used to test the effect of time (baseline vs. 12 weeks). All significant non-parametric analyses were adjusted for multiple comparisons via Bonferroni correction, since these analyses could not be run simultaneously. Composite Z-scores were computed for each cognitive construct by summing z-scores for each performance metric (e.g., accuracy, reaction time, learning rate, etc.) on a given task. All reaction times and error-based performance metric Z-scores were reverse scored so that positive values indicated better performance. The effects of the intervention on the detailed metrics of cognitive function that were used to compute these composite z-scores are displayed in the Appendix A. Effect sizes for our main effects are presented with their corresponding *p*-values and expressed as partial eta squared (η^2^) and Z/√n for ANOVA and non-parametric analyses, respectively. 

## 3. Results

Sample characteristics. Of the 122 adults originally recruited, 7 were lost to follow up prior to randomization, 16 dropped out of the trial, and 99 completed the 12-week intervention (Figure 1, *n* = 53 WPI, *n* = 46 CHO). Among the individuals who finished the trial (98% non-Hispanic white), WPI and CHO groups did not have statistically different (1) distribution of males and females, (2) prevalence of hypertension, dyslipidemia, asthma, and family history of CVD, (3) education status, and (4) depression (CES-D) and global cognitive function (MOCA) at baseline (Table 1). 

### 3.1. Intervention Effect on Anthropometrics and Secondary Outcomes

A main effect of time was observed for 6m walk, with participants walking slower at 12 weeks compared to baseline (*p* < 0.05; Table 2). A group-by-time interaction revealed BMI significantly increased from baseline to 12 weeks in the CHO group, with no mean changes observed for WPI. This effect was driven by a trend for increased body weight at 12 weeks in the CHO group compared to baseline (interaction effect *p* = 0.051). There were no significant main effects or interactions detected for blood lipids or glucose. 

### 3.2. Intervention Effect on Blood Pressure and Central Hemodynamics

The main effects of time were detected for brachial diastolic and mean pressure, with reductions at 12 weeks compared to baseline in both WPI and CHO (*p* < 0.05; Table 3). Significant group-by-time interactions were found for heart rate, aortic RPP, aortic stiffness (cfPWV), and aortic stiffness corrected for mean pressure (*p* < 0.05). Heart rate, aortic RPP (Figure 2B, Baseline: WPI 6965 ± 1088, CHO 6805 ± 1212; 12-week: WPI 6610 ± 1081, CHO 6794 ± 1191 mmHg/min), and aortic stiffness (Figure 2A, Baseline: WPI 10.1 ± 2.9, CHO 9.6 ± 2.5; 12-week: WPI 9.6 ± 2.7, CHO 10.1 ± 2.9 m/s) significantly decreased from baseline to 12 weeks in WPI, but not in CHO (*p* < 0.05). When expressed relative to mean pressure, aortic stiffness was unaltered in WPI but increased from baseline to 12 weeks in CHO (*p* < 0.05). No significant main effects or interactions were detected for pulse pressure, augmentation index, or aortic systolic pressure.

### 3.3. Intervention Effect on Cerebrovascular Hemodynamics

The main effects of time were detected for CCA and MCA mean blood velocity which decreased from baseline to 12 weeks in both WPI and CHO groups (*p* < 0.05; Table 4). No main effects were detected; however, for changes in MCA mean velocity relative to mean arterial pressure (i.e., MCA conductance). No significant main effects or interactions were detected for CCA and MCA pulsatility index, CCA forward or reflected wave intensity, and CCA diameter during the intervention. 

### 3.4. Intervention Effect on Cerebrovascular Response to Cognitive Activity

A main effect of time was detected for changes in MCA pulsatility during the Stroop task, which decreased more at 12 weeks compared to baseline in WPI and CHO (*p* < 0.05; Table 4). No main effects or interactions were observed for changes in CCA diameter, MCA mean velocity, or MCA conductance.

### 3.5. Intervention Effect on Cognitive Function

The main effects of time were observed for executive function and information processing composite scores, both of which increased (improved) from baseline to 12 weeks (*p* < 0.05; Table 5). A significant group-by-time interaction was detected for the emotion identification composite score which improved from baseline to 12 weeks in WPI but not CHO (*p* < 0.05). No significant main effects or interactions were detected for working memory, attention, memory, or impulsivity composite scores. 

## 4. Discussion

This study used a double-blind, randomized controlled trial to compare the effects of WPI versus CHO (control) on large artery stiffness, central blood pressure, cerebral responses to cognitive activity, and cognitive function in community dwelling older adults. Our results indicate that compared to 12 weeks of CHO supplementation, WPI supplementation resulted in modest reductions in aortic stiffness and central hemodynamic load (assessed as the product of aortic systolic pressure and heart rate). WPI had no effect on carotid vascular properties, cerebrovascular response to cognitive activity, and limited effects on cognitive function. Taken together, these data suggest that compared to CHO supplementation, WPI may favorably alter cardiovascular function in older adults but does not have a substantial impact on cerebrovascular or neurocognitive function. 

We noted reductions in aortic stiffness, assessed via gold-standard cfPWV, following 12 weeks of WPI compared to CHO. Our study adds to a growing literature noting favorable vascular effects of whey protein [23] with two RCTs discovering improved endothelial function with WPI in prehypertensive adults [22,38]. Aortic stiffness has been identified as a therapeutic target [13] owing to its ability to predict cardiovascular (CV) events and mortality [3,4], and offer insight into residual CVD risk [39]. The reduction in aortic stiffness with WPI may be partially driven by reductions in mean pressure. When expressed relative to blood pressure (cf PWV/MAP), we noted a slight increase in cfPWV in the CHO condition. This increase in arterial stiffness may be due to the effects of aging over the 12-week period, detrimental vascular effects of CHO, or both (i.e., CHO augmented age-associated increases in cfPWV and hastened vascular aging). WPI prevented increases in cfPWV/MAP seen with CHO. Even modest reductions in aortic stiffness are of physiological interest and clinically relevant. The effect of WPI on reducing aortic stiffness observed herein (~ −0.5 m/s) may help reduce CVD risk, particularly when compared with CHO (~ +0.5 m/s). A difference in aortic stiffness of 1.0 m/s is associated with a 15% reduction in CVD risk [3]. 

We observed significant reductions in aortic rate pressure product, a measure of central hemodynamic load with WPI compared to CHO. Rate pressure product is a mechanical hemodynamic parameter often used as proxy of cardiac oxygen consumption [40]. Aortic rate pressure product also quantifies the mechanical load experienced by the aorta (i.e., cycles of stretch) that contributes to aortic stiffening over time. Reductions in aortic rate pressure product herein appear driven by significant reductions in heart rate and modest reductions in aortic systolic pressure in the WPI versus CHO group. The unrelenting cyclic stress exerted by each cardiac contraction against the aortic walls amplifies oxidative stress and contributes to the fatigue and fracture of elastin [41], resulting in greater reliance on stiffer collagen fibers for wall load bearing. Indeed, a higher heart rate is associated with increased large artery stiffness [42]. Moreover, a higher resting heart rate is associated with accelerated progression of aortic stiffness over time [43]. Heart rate itself has been identified as an indicator and underappreciated risk factor for cardiovascular disease risk [44,45]. In addition to aforementioned links to large artery stiffening, elevated heart rate has been linked to inflammation, microalbuminuria, endothelial dysfunction, detrimental vascular remodeling and atherosclerosis [42]. Heart rate reduction has been shown in experimental (animal) studies to lower oxidative stress, restore endothelial function, and inhibit atherogenesis [42]. Reducing aortic rate pressure product via reductions in heart rate may thus help slow the progression of aortic stiffening over time by reducing cycles of stretch. Lower heart rate may also impact aortic stiffness via effects on mean (distension) pressure as a longer cardiac cycle with lower heart rate equates to more time spent in diastole. Overall, our findings suggest that compared to CHO, the heart rate-lowering effect of WPI in older adults may have a favorable effect on aortic stiffness and in turn CVD risk.

Given the significant association between increased heart rate and cardiovascular mortality, heart rate has been considered a potential therapeutic target [46]. Pharmacological means of lowering heart rate has been shown to reduce mortality in clinical populations [47] although this is not a universal finding across pharmacological agents. A meta-analysis of >68,000 patients suggests that lowering heart rate with Beta-blockers is associated with increased mortality risk in hypertensives [48]. This may be because the lowering heart rate with Beta-blockers is associated with increased pressure from wave reflections [49,50]. Heart rate is inversely related to pressure from wave reflections, such that lower heart rates are associated with increased pressure from wave reflections [51]. This inverse association may be stronger in older adults with higher aortic stiffness, such that a smaller decrease in heart rate may induce a larger increase in pressure from wave reflections [52]. Separate from cardiovascular effects of aortic stiffness, increased pressure from wave reflections increases cardiac afterload and is associated with numerous morbidities and mortality [3]. It is important to underscore that reductions in heart rate within the WPI group occurred without concomitant changes in global wave reflections, as augmentation index was unaltered. Reductions in HR with WPI may be related to ACE-inhibitory properties [53] but more research will be needed to further explore the potential mechanism. Thus, unlike what may be seen with select medications [54], WPI may be able to lower heart rate in older adults without having a detrimental impact on central hemodynamic load.

We noted no effects of WPI compared to CHO on carotid artery stiffness and subsequent cerebrovascular hemodynamics. Although the carotid artery and aorta are both considered central elastic arteries, they “age” at slightly different rates, are influenced differently by traditional CVD risk factors, and are differentially associated with target organ damage and cerebrovascular risk [55,56,57]. In the context of higher blood pressure, the aorta may stiffen more than the carotid artery, suggesting the higher sensitivity of the aorta to the effects of aging [57]. It is possible that each artery may respond differently to dietary manipulation and that the aorta may be more sensitive to supplementation with WPI compared to the carotid artery. When performing secondary exploratory analyses, central hemodynamic load (RPP) was associated with cfPWV at baseline (*r* = 0.38, *p =* 0.001) and 12 weeks (*r* = 0.25, *p* = 0.015) and change in RPP across the intervention was associated with change in cfPWV (*r* = 0.38, *p* = 0.007). Conversely, central hemodynamic load (RPP) was not associated with carotid stiffness at baseline (*r* = 0.09, *p =* 0.39) and 12 weeks (*r* = −0.04, *p =* 0.71), and change in RPP across the intervention was not associated with change in carotid stiffness (*r* = 0.09, *p =* 0.40). The carotid artery may be more resilient to the effects of hemodynamic load. Overall, our findings suggest that although lowering central hemodynamic load with WPI may have a favorable effect on aortic stiffness compared to CHO, this does not translate to a similar favorable effect on carotid artery stiffness. 

Cerebrovascular and cognitive function was generally unaltered with WPI compared to CHO. We did observe a select improvement in emotion identification among the WPI group compared to CHO, which is linked to general cognition and may play an integral role in the organization of information processing [58]. Despite this modest effect, we noted no further significant group differences in executive function, memory/working memory, attention, impulsivity, or information processing. The latter findings are consistent with a recent meta-analysis suggesting that milk and dairy intake may not improve cognitive function in older adults [59], and an RCT noting that WPI specifically may not impact cognitive function [60]. Overall, it is possible that the common carotid artery serves as an extra-cranial “gatekeeper”, either buffering entry of hemodynamic pulsatility to the brain or facilitating transmission of excess pulsatile energy [61]. Thus, by not affecting carotid artery stiffness, WPI may have been ineffective in further impacting cerebrovascular response to cognitive activity and cognitive function itself. 

We chose maltodextrin as our control, given its similar appearance and energy yield to WPI and we view this as a delimitation. Maltodextrin is the most commonly used control reported in this area of scholarship. Maltodextrin may not be an inert placebo and could have influenced some outcome measures. Although maltodextrin is a polysaccharide and complex CHO, when compared to WPI, maltodextrin has a higher glycemic index [62] which in theory could have a detrimental effect on glycemic control and thus vascular function. As mentioned previously, our findings should be interpreted in the context that noted interactions (cfPWV) could have been driven partially by a) direct detrimental effects of CHO on vascular function and thus aortic stiffness, and/or b) CHO amplifying vascular aging. We believe this unlikely as a recent ancillary study from the large multicenter Protein Supplementation Trial noted specifically that although maltodextrin supplementation (45 g/daily for 18 months) increases glycemic load, it does not have a detrimental effect on systemic inflammation or insulin resistance [63]. Moreover, maltodextrin does not have detrimental effects on vascular endothelial function measured as brachial flow-mediated dilation or circulating biomarkers of vascular endothelial inflammation [22,38]. Our study noted no detrimental effects of maltodextrin on blood pressure, heart rate or other cardiometabolic parameters (e.g., fasting glucose and triglycerides), all notable determinants of change in PWV over time [64,65,66]. Thus while maltodextrin is not a completely “inert placebo”, it does not appear to have effects on many of the more prominent determinants of vascular stiffness (i.e., systemic inflammation, vascular inflammation, vascular endothelial function, blood pressure, heart rate, insulin resistance). 

We did not control participants’ dietary intake, and this may be viewed as a study limitation. The majority of previous studies comparing WPI to CHO have found expected changes in macronutrient composition when comparing supplement groups (i.e., higher dietary protein composition and lower CHO in the WPI group and higher dietary CHO composition with lower protein in the CHO group) [22,38,67,68,69]. Interestingly, although changes in macronutrients occur as expected, older adults may decrease ad libitum caloric intake resulting in an iso-caloric total energy state across the intervention period [70]. Increased intake of one macronutrient results in a decreased intake of at least one of the other macronutrients under isocaloric conditions [71]. Thus, we cannot distinguish the effects of increased protein consumption from the effects of possible decreased carbohydrate consumption in the WPI group. WPI may have a slightly greater effect on satiety resulting in an anorexigenic response compared to maltodextrin [62], and a slightly greater reduction in total caloric intake across the intervention compared to CHO [72]. Our results would support these observations as there was no significant change in body mass in the WPI group but a small nonsignificant increase (~1 kg) in the CHO group. Of overall importance for the interpretation of our findings, statistically adjusting for change in BMI did not have an effect on group differences in cfPWV thus it is unlikely that the ~1kg increase in body mass in the CHO group was responsible for noted group differences in aortic stiffness. 

Additional limitations should be noted. Our participants were well-educated with ~75% completing college and an additional ~40% completing a graduate degree. Education may increase cognitive reserve and be related to a more enriched environment later in life, preserving cognitive flexibility [73]. As such, our participants were a higher functioning group with less room for cognitive improvement. Our study population comprised 98% non-Hispanic white adults. There are known racial differences in large artery stiffness [74] and the blood pressure response to ACE-inhibitors [75]. Additional research exploring the impact of WPI on cardiovascular health in non-Hispanic black/African American adults and Hispanic adults is warranted. Finally, our strict exclusion criteria may limit the generalizability of our findings to other populations of older adults (e.g., diabetics). 

## 5. Conclusions 

In conclusion, WPI has a modest but favorable effect on aortic stiffness and central hemodynamic load (appraised as the product of heart rate and central systolic blood pressure) in community-dwelling older adults when compared to a CHO control. WPI may not affect carotid artery stiffness or cerebrovascular response to cognitive activity, and appears to have limited effects on cognitive function among older adults. Although a logical extension of our findings would suggest WPI has potential as a nutritional strategy to help manage age-associated increases in aortic stiffness, further studies utilizing longer intervention periods and alternative control conditions will be needed to corroborate our findings.

## Figures and Tables

**Figure 1 nutrients-12-01054-f001:**
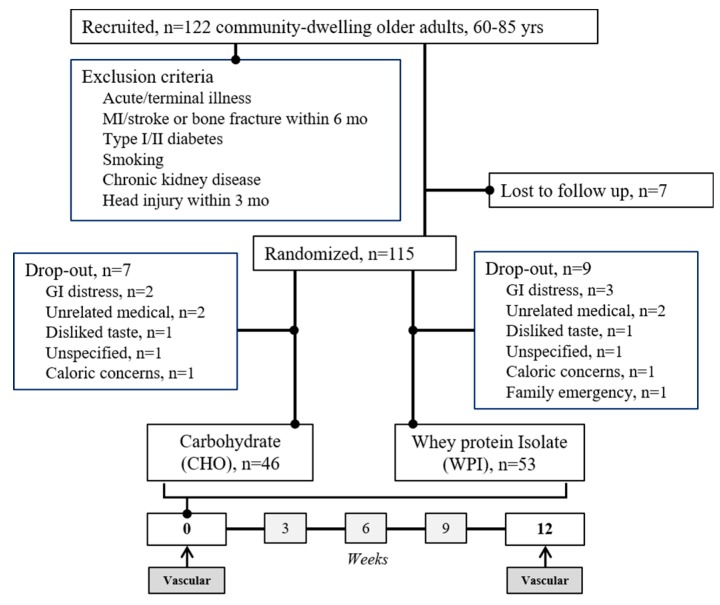
Participant exclusion criteria and recruitment/enrollment flow chart. MI, myocardial infarction; mo, months; yrs, years; GI, gastrointestinal.

**Figure 2 nutrients-12-01054-f002:**
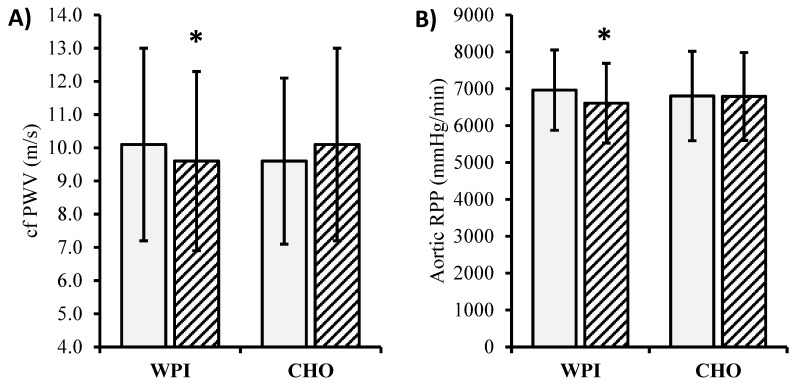
Changes in (**A**) aortic stiffness (cf PWV) and (**B**) aortic rate pressure product (RPP) from baseline to 12 weeks in WPI (whey protein isolate) vs. CHO (carbohydrate) (mean ± SD). Effects, *p*-value (partial η^2^). * *p* < 0.05 vs. baseline. (A) Group 0.88(0.00); time 0.88(0.00); G × T **0.01(0.00)**; df 96. (B) Group 0.96(0.00); time **0.03(0.05)**; G × T **0.04(0.04)**; df 94. Bold highlights statistically significant effects.

**Table 1 nutrients-12-01054-t001:** Descriptive characteristics (*n* (%), unless otherwise noted).

	WPI *n* = 53	CHO *n* = 46	*p* Value	df
Female Sex	26(49.1)	19(41.3)	0.54	98
Asthma	1(1.9)	0(0.0)	1.00	98
Hypertension	23(43.4)	17(37.0)	0.54	98
Dyslipidemia	23(43.4)	28(60.9)	0.11	98
Fam Hx CVD	32(60.4)	28(62.2)	1.00	98
Handedness			0.25	98
Right	42(79.2)	40(87.0)		
Left	9(17.0)	3(6.5)		
Ambidextrous	2(3.8)	3(6.5)		
Education			0.08	98
High School	5(9.4)	1(2.2)		
Some college	8(15.1)	3(6.5)		
2-yr Degree	3(5.7)	1(2.2)		
College	12(22.6)	21(45.7)		
Graduate degree	25(47.2)	20(43.5)		
Age (years) ^	69 ± 7	67 ± 6	0.25	98
Height (m) ^	1.67 ± 0.13	1.68 ± 0.10	0.95	98
CESD Score ^	7 ± 6	7 ± 8	0.89	95
MOCA Score ^	27 ± 2	27 ± 2	0.70	98

^ mean ± SD. WPI, whey protein isolate; CHO, carbohydrate; Fam Hx CVD, Family history of cardiovascular disease; CESD, Center for Epidemiological Studies Depression; MOCA, Montreal Cognitive Assessment; df, degrees of freedom.

**Table 2 nutrients-12-01054-t002:** Changes in anthropometrics, blood lipids, mobility, and global cognition from baseline to 12 weeks in WPI vs. CHO (mean ± SD).

	WPI		CHO		Effects, *p*-Value (Partial η^2^)			
	Baseline	12 weeks	Baseline	12 weeks	Group	Time	Group × Time	df
Weight (kg)	78.2 ± 16.3	78.1 ± 16.3	76.3 ± 14.5	77.3 ± 15.0	0.67(0.00)	0.18(0.02)	0.051(0.04)	98
BMI (kg/m^2^)	27.9 ± 5.6	27.8 ± 5.6	27.0 ± 3.9	27.4 ± 4.1 *	0.65(0.00)	0.18(0.02)	**0.04(0.04)**	98
Body fat (%)	30.9 ± 12.3	29.8 ± 13.3	29.4 ± 9.9	29.8 ± 10.7	0.76(0.00)	0.51(0.01)	0.14(0.02)	93
Waist Circ (cm)	96.6 ± 13.2	96.1 ± 13.8	95.8 ± 12.0	96.3 ± 12.5	0.91(0.00)	0.97(0.00)	0.32(0.01)	94
Total cholesterol (mg/dL)	189 ± 34	184 ± 36	182 ± 33	183 ± 34	0.51(0.01)	0.41(0.01)	0.20(0.02)	93
HDL (mg/dL)	60 ± 22	58 ± 21	57 ± 19	55 ± 19	0.53(0.00)	0.11(0.03)	0.82(0.00)	92
LDL (mg/dL)	105 ± 30	103 ± 34	98 ± 27	101 ± 26	0.39(0.01)	0.80(0.00)	0.36(0.01)	83
Triglycerides (mg/dL)	112 ± 46	107 ± 48	129 ± 78	133 ± 57	0.08(0.03)	0.70(0.00)	0.11(0.03)	91
Glucose (mg/dL)	94 ± 14	95 ± 15	92 ± 11	94 ± 11	0.43(0.01)	0.09(0.03)	0.53(0.00)	93
6 m Walk (s)	4.72 ± 0.64	4.78 ± 0.72	4.59 ± 0.64	4.86 ± 0.71	0.80(0.00)	**<0.01(0.08)**	0.07(0.04)	88
IPAQ ^	3065 ± 3023	2602 ± 2552	4597 ± 4009	4367 ± 4314	0.15(0.22)	0.32(0.18)	0.99(0.02)	78
MVPA (min/d) ^	23.2 ± 19.3	19.2 ± 15.8	26.0 ± 27.6	30.9 ± 34.0	0.99(0.08)	0.99(0.04)	0.13(0.25)	65

^ Non-parametric analyses with Bonferroni correction, effect sizes calculated as Z/√n. WPI, whey protein isolate; CHO, carbohydrate; BMI, body mass index; HDL, high density lipoprotein; LDL, low density lipoprotein; IPAQ, international physical activity questionnaire; MVPA, moderate-to-vigorous physical activity. * *p* < 0.05 vs. Baseline. Bold highlights statistically significant effects.

**Table 3 nutrients-12-01054-t003:** Changes in blood pressure and central hemodynamics from baseline to 12 weeks in WPI vs. CHO (mean ± SD).

	WPI		CHO		Effects, *p*-Value (Partial η^2^)			
	Baseline	12 weeks	Baseline	12 weeks	Group	Time	Group × Time	df
Brachial								
Systolic pressure (mmHg)	125 ± 13	123 ± 12	127 ± 11	128 ± 12	0.10(0.03)	0.42(0.01)	0.13(0.02)	96
Diastolic pressure (mmHg)	79 ± 8	76 ± 7	79 ± 5	78 ± 6	0.30(0.01)	**<0.01(0.09)**	0.20(0.02)	96
Pulse pressure (mmHg)	46 ± 9	46 ± 9	48 ± 8	49 ± 9	0.13(0.02)	0.28(0.01)	0.28(0.01)	96
Mean pressure (mmHg)	94 ± 9	92 ± 8	95 ± 7	95 ± 7	0.12(0.03)	**0.03(0.05)**	0.13(0.02)	96
Aorta								
Augmentation index 75	24 ± 10	23 ± 11	25 ± 9	25 ± 8	0.25(0.01)	0.50(0.01)	0.52(0.00)	96
Systolic pressure (mmHg)	115 ± 13	112 ± 12	117 ± 11	119 ± 12	**0.04(0.04)**	0.48(0.00)	0.09(0.03)	97
Pulse pressure (mmHg)	35 ± 10	35 ± 7	37 ± 8	39 ± 10	0.06(0.04)	0.24(0.01)	0.26(0.01)	97
Heart rate (b/min)	60 ± 8	56 ± 8 *	57 ± 9	56 ± 9	0.39(0.01)	**<0.001(0.13)**	**<0.01(0.07)**	96
Cf-PWV/MAP (m/s/mmHg × 10^2^)	10.7 ± 3.1	10.5 ± 3.0	10.0 ± 2.3	10.6 ± 2.8 *	0.73(0.00)	0.46(0.01)	**0.03(0.05)**	95

WPI, whey protein isolate; CHO, carbohydrate; Cf-PWV/MAP, carotid femoral pulse wave velocity relative to mean arterial pressure; * *p* < 0.05 vs. Baseline. Bold highlights statistically significant effects.

**Table 4 nutrients-12-01054-t004:** Cerebrovascular hemodynamics at rest and in response to cognitive activity (∆Stroop) at baseline and 12 weeks in WPI vs. CHO (mean ± SD).

	WPI		CHO		Effects, *p*-Value (Partial η^2^)			
	Baseline	12 weeks	Baseline	12 weeks	Group	Time	Group × Time	df
Common Carotid Artery								
Pulsatility index	1.37 ± 0.24	1.40 ± 0.30	1.40 ± 0.28	1.41 ± 0.29	0.77(0.00)	0.34(0.01)	0.84(0.00)	96
β-stiffness	8.5 ± 4.3	8.5 ± 3.8	8.0 ± 2.5	9.3 ± 5.0	0.39(0.01)	0.45(0.01)	0.16(0.02)	94
W1 (mmHg/m/s^3^)	7.1 ± 4.7	7.2 ± 4.1	7.2 ± 4.2	7.6 ± 6.6	0.84(0.00)	0.91(0.00)	0.75(0.00)	93
NA (mmHg/m/s^2^)	22.9 ± 11.6	24.5 ± 14.4	25.2 ± 16.6	26.0 ± 23.8	0.80(0.00)	0.80(0.00)	0.34(0.01)	93
Mean Diameter (mm)	5.92 ± 0.62	5.88 ± 0.63	5.86 ± 0.65	5.86 ± 0.67	0.48(0.01)	0.55(0.00)	0.48(0.01)	96
∆Diameter (mm)	+0.11 ± 0.15	+0.17 ± 0.15	+0.10 ± 0.20	+0.13 ± 0.13	0.08(0.03)	0.31(0.01)	0.57(0.00)	95
IMT (mm)	0.66 ± 0.10	0.68 ± 0.12	0.67 ± 0.14	0.67 ± 0.14	0.80(0.00)	0.24(0.01)	0.09(0.03)	96
Mean velocity (cm/s)	54 ± 12	52 ± 14	53 ± 15	51 ± 14	0.85(0.00)	**<0.01(0.09)**	0.65(0.00)	92
Middle Cerebral Artery								
Mean velocity (cm/s)	54 ± 12	52 ± 14	53 ± 15	51 ± 14	0.85(0.00)	**<0.01(0.09)**	0.65(0.00)	93
∆Mean velocity (cm/s)	+5 ± 7	+4 ± 5	+5 ± 6	+5 ± 5	0.38(0.01)	0.36(0.01)	0.57(0.00)	92
Pulsatility index	0.86 ± 0.15	0.88 ± 0.18	0.87 ± 0.13	0.88 ± 0.18	0.79(0.00)	0.67(0.00)	0.62(0.00)	93
∆Pulsatility index	+0.00 ± 0.06	−0.02 ± 0.05	−0.00 ± 0.05	−0.01 ± 0.05	0.70(0.00)	**0.02(0.06)**	0.42(0.01)	92
Conductance (cm/s/mmHg × 10^2^)	58.0 ± 15.2	57.2 ± 17.0	55.6 ± 16.8	54.5 ± 17.1	0.54(0.00)	0.13(0.02)	0.77(0.00)	93
∆Conductance (cm/s/mmHg × 10^2^)	+0.2 ± 6.0	−0.9 ± 6.6	+0.2 ± 6.5	+0.4 ± 5.6	0.51(0.01)	0.60(0.00)	0.41(0.01)	90

WPI, whey protein isolate; CHO, carbohydrate; IMT, intima media thickness; NA, negative area; ∆denotes change in variable from rest to during cognitive Stroop perturbation. Bold highlights statistically significant effects.

**Table 5 nutrients-12-01054-t005:** Composite Z-scores across cognitive function domains in WPI vs. CHO at baseline and 12 weeks (mean ± SD).

		WPI		CHO		Effects, *p*-Value (z/sqrt(*n*))			
Domain/Construct	Task	Baseline	12 weeks	Baseline	12 weeks	Group	Time	Group × Time	df
Executive function	Maze	−0.34 ± 2.80	0.34 ± 2.14	−0.29 ± 2.08	0.74 ± 2.62	0.99(0.07)	**<0.01(0.33)**	0.99(0.08)	93
Impulsivity	Go-no-go	−0.23 ± 1.65	0.28 ± 1.16	−0.12 ± 1.54	0.18 ± 1.12	0.99(0.04)	0.051(0.25)	0.77(0.12)	92
Emotion Identification	Recognition	−0.14 ± 1.15	0.43 ± 1.37 *	−0.15 ± 1.98	−0.11 ± 2.06	0.87(0.11)	**<0.01(0.31)**	**0.04(0.25)**	96
Memory	Recall	0.30 ± 5.76	0.66 ± 5.25	−0.21 ± 5.29	−0.43 ± 6.40	0.53(0.14)	0.99(0.05)	0.36(0.16)	94
Information Processing	Interference	0.85 ± 4.73	2.30 ± 3.62	−0.06 ± 5.75	0.62 ± 5.19	0.98(0.46)	**0.045(0.27)**	0.92(0.11)	81
Attention	CPT	0.02 ± 2.05	0.03 ± 1.82	−0.10 ± 2.34	0.28 ± 1.56	0.99(0.04)	0.79(0.11)	0.77(0.12)	95
Working memory	Digit span	0.23 ± 1.65	0.27 ± 1.93	−0.38 ± 2.23	0.00 ± 1.71	0.52(0.09)	0.99(0.09)	0.99(0.05)	87

WPI, whey protein isolate; CHO, carbohydrate; CPT, continuous performance test; * *p* < 0.05 vs. Baseline. Bold highlights statistically significant effects.

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
