# Peer review of "Effects of Whey Protein Supplementation on Aortic Stiffness, Cerebral Blood Flow, and Cognitive Function in Community-Dwelling Older Adults: Findings from the ANCHORS A-WHEY Clinical Trial"

_nutrients, 2020, doi:10.3390/nu12041054_

Round 1
Reviewer 1 Report
This study was a 12-week trial comparing whey protein supplementation to a carbohydrate (maltodextrin) supplement. The authors found a slight reduction in arterial stiffness with the whey protein supplement, and at the same time, the carbohydrate supplement group had increased arterial stiffness. There were not any interactions between treatment group and time for blood pressure measures. Aortic rate pressure product was reduced in the whey protein group, and this could be driven by a reduced heart rate which was also significant in the whey protein group. There were not any significant differences in measures of cerebrovascular hemodynamics and only one (of seven) cognitive function tests had a significant treatment interaction.
This study assessed a multitude of hemodynamic and cognitive outcomes using gold standard techniques. Overall, there was careful attention to confounding factors and a strong selection criteria for subjects; however, there is a potential confound of using a control group that had maltodextrin supplementation. For this and other reasons described below, it is suggested that the conclusions of the study be tempered.
Major comments:
-The use of maltodextrin as the control supplement adds a level of confounding factors. Due to this control group, it unclear if the treatment interactions actually result from the whey protein, or are a difference in response to protein vs. carbohydrate supplement. For example, the increase in PWV in the control group could be a normal age-related change or it could be a negative effect of the maltodextrin supplementation.
-Overwhelmingly, the results of the study are negative. It is still important to publish these results, but the focus on the very (physiologically) small changes that are significant is somewhat misleading. For this reason and the issue of the control group (see above), the conclusions/discussion need revising to realistically reflect the results of the study.
-The dietary intake of the subjects during the intervention are important data that are not presented. If either supplement group changed their overall caloric or macronutrient intake during the intervention period, then this could have a confounding effect on the results. For example, did the whey protein group replace protein intake with the supplement, or did they have an overall increase in protein intake, with perhaps a reduced intake of other macronutrients.
-The terminology for “cerebrovascular reactivity” should be reconsidered. MCA blood velocity response to a cognitive task could be affected by both the cerebrovascular function and differences between subjects in the neuronal activity. In contrast, typical measures of cerebrovascular reactivity in response to a stimulus, such as hypercapnia, are more influenced by only differences in vascular function between subjects.
-A discussion of the limitation of the strict subject exclusion criteria and the effects on the generalizability of the results should be considered.
Minor comments:
-The calculation for RPP needs to be included in the Methods.
-There are inconsistencies in the tables for the bolding/significance markers.
-Line 300 is missing a comma
-Line 414: “invention” vs “intervention”
Reviewer 2 Report
The manuscript entitled “Effects of Whey Protein Supplementation on Aortic Stiffness, Cerebral Reactivity and Cognitive Function in Community-Dwelling Older Adults: Findings from the ANCHORS A-WHEY Clinical Trial” sought to assess the effect of a 12-week whey protein intake on large artery stiffness, cerebral reactivity and cognitive function in older adults. The authors found that whey protein isolate (WPI) supplements result in favorable reductions in aortic stiffness (cfPWV) and aortic hemodynamic load (SBPxHR) with limited effects on cognitive function (emotion recognition) and no effect on cerebrovascular function in community-dwelling older adults. The manuscript was well written in general and data analysis was thorough and detailed. Below are comments that will further strengthen the study:
1) Page 10, Line 201, Figure 2A should be Figure 2B, and Line 302, Figure 2B should be Figure 2A.
2) In Figure 2A (as the authors presented), the baseline for WPI and CHO appears to be different. The author should address this issue.
3)50g of carbohydrates are considered as a very low daily carbohydrate intake. Was there any dietary restriction on participant’s daily food intake, energy expenditure (exercise/training)? The authors should clarify it.
4) A statistically significant effect was seen within 12 weeks of intervention is very promising. What is the expected long-term significance? Any follow-up study? What about a higher intake WPI amount? These questions all lead to one basic question that needs to be address by the authors: What was the rationale/reference in using the dosage of 50g WPI?
